# Synthesis, Self-Assembly and Characterization of Tandem Triblock BPOSS-PDI-X Shape Amphiphiles

**DOI:** 10.3390/molecules24112114

**Published:** 2019-06-04

**Authors:** Yu Shao, Jia Chen, Xiang-Kui Ren, Xinlin Zhang, Guang-Zhong Yin, Xiaopeng Li, Jing Wang, Chrys Wesdemiotis, Wen-Bin Zhang, Shuguang Yang, Bin Sun, Meifang Zhu

**Affiliations:** 1State Key Laboratory for Modification of Chemical Fibers and Polymer Materials and College of Material Science and Engineering, Center for Advanced Low-Dimension Materials, Donghua University, Shanghai 201620, China; boukephalas@outlook.com (Y.S.); 2170464@mail.dhu.edu.cn (J.C.); zhangxl2015@sjtu.edu.cn (X.Z.); 2School of Chemical Engineering and Technology, Tianjin University, Tianjin 300350, China; renxiangkui@tju.edu.cn; 3Department of Chemistry, Key Laboratory of Polymer Chemistry and Physics of Ministry of Education, Center for Soft Matter Science and Engineering, College of Chemistry and Molecular Engineering, Peking University, Beijing 100871, China; gzh_yin@126.com (G.-Z.Y.); wenbin@pku.edu.cn (W.-B.Z.); 4Department of Chemistry, University of South Florida, 4202 East Fowler Ave, Tampa, FL 33620, USA; xiaopengli1@usf.edu; 5South China Advanced Institute of Soft Matter Science and Technology, South China University of Technology, Guangzhou 510640, China; wangj81@scut.edu.cn; 6Department of Chemistry, The University of Akron, Akron, OH 44325, USA; wesdemiotis@uakron.edu

**Keywords:** shape amphiphile, self-assembly, nano-belt single crystal, PDI, POSS

## Abstract

In this article, we report the facile synthesis, self-assembly, and characterization of shape amphiphiles (BPOSS-PDI-X) based on isobutyl-functionalized polyhedral oligomeric silsesquioxane (BPOSS), perylene tetracarboxylic diimide (PDI), and (60)fullerene (C_60_) moieties. Firstly, an asymmetrically functionalized diblock shape amphiphile precursor (BPOSS-PDI-OH) was obtained through the one-pot reaction between perylene-3,4,9,10-tetracarboxylic dianhydride and two different amines, namely BPOSS-NH_2_ and 3-amino-1-propanol. It was further conjugated with C_60_-COOH to give a tri-block shape amphiphile (BPOSS-PDI-C_60_). Their chemical structures were thoroughly characterized by NMR, IR and MALDI-TOF MS spectrometry. In order to gain insights on the structure-property relationship, their self-assembly in gas phase, in solution, and in solid state were characterized using traveling wave ion mobility mass spectrometry (TWIM-MS), UV/Vis absorption, fluorescence emission spectrophotometer, and transmission electron microscopy, respectively. It was found that BPOSS-PDI-OH formed more complicated dimers than BPOSS-PDI-C_60_. Both samples showed unique aggregation behaviors in solution with increasing concentration, which could be attributed neither to H- nor to J-type and might be related to the discrete dimers. While BPOSS-PDI-C_60_ could hardly crystalize into ordered structures, BPOSS-PDI-OH could form nanobelt-shaped single crystals, which may hold potential applications in microelectronics.

## 1. Introduction

Echoing the Material Genome Initiative [1], the study of shape amphiphiles has brought us a new way to understanding structure-property relationships in materials science by combining molecular moieties of different shapes and functions and letting anisotropy promote their hierarchical self-assembly [2,3,4]. Shape amphiphiles are built using shape-persistent molecular nanoparticles with rigid configurations [5,6,7,8,9]. For example, polyhedral oligomeric silsesquioxanes (POSS) are typical cage-like molecules and are also known as organic-inorganic hybrid materials [10,11]. (60)Fullerene (C_60_) is an organic shape-persistent molecule with spherical skeleton possessing unique π–π interactions and intriguing electronic properties [12,13,14,15,16,17]. Using these molecular nanoparticles, one can build shape amphiphiles with uniform molecular size and precise molecular structures [18]. These amphiphilic molecules can be chemically tailored like small molecules, but also possess fixed, rigid three-dimensional molecular shapes like particles that do not easily deform. For the simple sphere-cube shape amphiphile like the BPOSS-C_60_ conjugate (where BPOSS stands for isobutyl-functionalized POSS), polymorphism has been observed, both of which share the double-layered packing scheme with alternating, phase-separated POSS and C_60_ structures [19]. BPOSS-C_60_ has also been identified as an excellent electron acceptor for inverted bulk heterojunction polymer solar cells [20]. In fact, the introduction of shape anisotropy has been shown to be extremely versatile in creating unconventional phases including the Frank-Kasper and dodecagonal quasicrystal phases in soft matters [21,22,23,24,25].

Adding a third block to the shape amphiphile would be anticipated to generate even more complexity and diversity [26]. For example, tandem triblock shape amphiphilic regio-isomers based on POSS were found to exhibit distinct self-assembling behaviors [27]. If the third block is of distinct shape, the situation would be even more interesting. Being one of the most popular planar molecular motifs, perylene tetracarboxylic diimide (PDI) has attracted intense research interests over the past decades. It exhibits interesting photo- and electro-properties such as high charge carrier mobility [28,29,30], high fluorescent quantum efficiency for cell imaging [31], high performance in all-polymer solar cells [32,33,34], readily tunable emission spectra [35], etc. The strong π–π interactions also make it a robust and strongly self-assembling motif both in solution and in bulk [36,37,38,39,40,41,42,43]. A PDI-C_60_ conjugate was reported as the light-harvesting dyad [44]. However, significant quenching with very low quantum efficiency has been commonly observed for PDI derivatives in solid state, which hampers practical applications [45]. Recently, it was demonstrated that the introduction of bulky side groups can break the close packing of PDI into discrete aggregates and improve the fluorescent properties [46,47,48,49]. For example, the fluorescence quantum yields (*Φ_f_*) of PDI in the crystalline state can be as high as 48% for BPOSS-PDI-BPOSS which is a PDI tethered with two bulky BPOSS motif at the imide position [50]. Incorporating the PDI motif into the POSS-C_60_ conjugate is thus very intriguing, especially in the fields of electro-/photo- devices. On one hand, attaching the BPOSS motif to PDI increases the solubility and reduces the π–π interaction due to the steric hindrance of the bulky neighboring BPOSS. On the other hand, the interplay between different motifs might lead to ordered arrangement of PDI and C_60_ to take advantage of their useful electronic properties [51]. In this case, we anticipate that the assembly of the triblock shape amphiphile could help control the molecular packing and crystal morphology of the assemblies, which is critical to their applications in nanostructure engineering [52,53] and single crystal devices [54,55]. In this contribution, we report the synthesis, self-assembly, and characterization of two unsymmetrical shape amphiphiles, BPOSS-PDI-OH and BPOSS-PDI-C_60_. These model compounds were thoroughly characterized in terms of their assembly in gas phase, in solution, and in condensed state.

## 2. Results and Discussion

### 2.1. Molecular Design and Synthesis

As shown in Scheme 1, BPOSS-PDI-OH was synthesized via a one-pot condensation method where the two amines were competing to react with the dianhydride to form the diadducts [56]. The reaction conditions were optimized to get the desired asymmetry diadduct as much as possible. Firstly, half of the BPOSS-NH_2_ (0.5 equivalent) was added to PTCDA (1 equivalent) in imidazole in the three-necked round bottom flask and the mixture was stirred at 140 °C. Then, the other half of BPOSS-NH_2_ (0.5 equivalent) along with 3-amino-1-propanol (0.5 equivalent) in THF were added slowly to the above mixture using syringe pump. Using this protocol, BPOSS-PDI-OH were obtained in an acceptable yield of ~30%. Following the Steglich esterification with fullerene acid (C_60_-COOH), the final cubic-plane-sphere triblock shape amphiphile (BPOSS-PDI-C_60_) was obtained in excellent yield (~87%). The chemical identity and purity of the two products were thoroughly characterized by ^1^H-NMR, ^13^C-NMR, FT-IR and MALDI-TOF MS spectrometry.

Figure 1 shows their ^1^H-NMR spectra. The characteristic peaks of the aromatic protons of PDI can be clearly seen around 8.6 ppm (a, b, a’, b’). Due to the high asymmetry of BPOSS-PDI-OH, the protons adjacent to imide group were split into two distinct groups whose peaks were labelled as c (4.41 ppm) and c’ (4.21 ppm), respectively. The protons near to hydroxyl group have a much higher chemical shift (e’, 3.66 ppm) compared to the protons near the silicon atom (e, 0.74 ppm). After esterification, the signal for the proton near the hydroxyl group (e’) shifts from 3.66 ppm to 4.22 ppm completely. The incorporation of strongly electron-withdrawing C_60_ motif is corroborated by the downfield shift of the resonances at d’, c, and c’. The corresponding ^13^C-NMR spectra are shown in Appendix A (see Appendix A). After successful conjugation with C_60_, there appear multiple peaks at around 142 ppm that could be assigned to *sp*^2^ carbons on C_60_. From FT-IR spectra, we can also see a sharp peak at 524 cm^−1^ in BPOSS-PDI-C_60_ that is characteristic of the F_1u_ vibration of C_60_ skeleton (Appendix A). To further prove the chemical identity of these compounds, we carefully run the MALDI-TOF MS experiment for each sample under the reflection mode. As shown in Appendix A, the monoisotopic peak could be clearly detected for both compounds and the observed molecular weight matches well with that of the calculated values. For example, the BPOSS-PDI-OH that ionizes with Na^+^ appears at *m*/*z* of 1327.6, which is consistent with the calculated monoisotopic mass of 1327.5 Da (Appendix A). The observed *m*/*z* of BPOSS-PDI-C_60_ (2087.4 *m*/*z*) also matches well with the calculated value at 2087.4 Da (Appendix A). We further investigated the thermal stability and phase behaviours of the two shape amphiphiles using thermogravimetric analysis (TGA) and differential scanning calorimetry (DSC). From the thermograms, it could be seen that both samples have very good thermal stability with the 5% weight loss temperature well above 300 °C (Appendix A). There is no phase transition from −10 °C to 200 °C (Appendix A). The results suggest that these shape amphiphiles are excellent model compounds for further self-assembly studies.

### 2.2. Self-Assembly in Gas Phase as Revealed by ESI-TWIM-MS Spectrometry

ESI-TWIM-MS spectrometry is a powerful technique for studying noncovalent interactions [57]. The principle is based on the fact that ions exhibit different mobility when traveling through an inert gas under electric field. The mobility will be dependent on various factors including the strength of electric fields, gas flow, shape, size, and net charge of the molecular ions. The technique is sensitive in distinguishing various conformers (molecules/complexes) with the same *m*/*z* values. Information about the molecular conformation or aggregation can be extracted from the ESI-TWIM-MS spectrometry for both biological and nonbiological molecules [58,59,60]. Moreover, the stability of the complex can be further probed by applying collisional energy. All these features have made ESI-TWIM-MS spectrometry a versatile and powerful platform for the characterization of supramolecular assemblies. The method is thus applied here for studying the aggregation states of PDI molecules. Planar PDI molecules tend to form extended aggregates with continuous stacking of π planes in most cases. However, when the side chains are bulky, the steric hindrance may result in discrete packing using dimer motifs rather than continuous packing. Since BPOSS and C_60_ are molecular nanoparticles with distinct shapes, their combination is expected to impart a considerable constraint on the packing of the planar PDI. The immiscibility between BPOSS and C_60_ and the interplay between different secondary interactions would promote the hierarchical assembly further toward ordered structure formation. To achieve a free energy minimum, molecular conformational reorganization would occur with longitudinal, transverse, and rotational offsets between neighboring PDI [50]. We used ESI-TWIM-MS spectrometry to reveal the aggregation states of the PDI derivatives and characterize their relative stability by disrupting them with collisionally activated dissociation (CAD). It was previously found that when the side chains are bulky as in BPOSS-PDI-BPOSS, dimers were the most predominant form of aggregates and also the most favored motif for further packing into crystals [46]. It is intriguing to see how these asymmetrically tethered PDIs would behave.

The full TWIM-MS spectrum of BPOSS-PDI-C_60_ shows that there are monomers, dimers, trimers, and higher oligomers (Figure 2A). The strongest peak at *m*/*z* = 2087 can be attributed to either the monomer or dimer peaks found in the form of [M∙Na^+^] or [2M∙2Na^+^], while trimers [3M∙Na^+^] and tetramers [4M∙Na^+^] were only detected with trace abundance. A zoom-in view of the dimers at *m*/*z* = 2087 in Figure 2A (inset) reveals that there are mainly three types of dimers (namely, dimers **1**–**3**) as resolved by TWIM. Then, tandem mass spectrometry was used in the trap cell to break the aggregation. The applied collision energy (*E*_CM_) was varied from 0.11 to 0.70 eV. Both dimers **1** and **2** could be dissociated with an applied *E*_CM_ of 0.38 eV and the signal of monomer increases correspondingly (Figure 2B). However, the signal of dimer **3** persists up to the *E*_CM_ of 0.70 eV. The results indicate that compared to dimers **1** and **2**, the dimer **3** is bound together more tightly and higher energy is required to dissociate it. The strong binding in dimer **3** is probably attributed to the strong π–π interactions between C_60_ moieties that require much higher energy to break. Dimers **1** and **2** dissociate at a lower energy level, implying that they are only weakly bound together. However, the details of their interaction patterns remain to be illustrated.

To reveal more information about their molecular interactions, we studied two model compounds to gain information about the energy required to break the π–π interaction between C_60_ motifs. The two referential shape amphiphiles are BPOSS-C_60_ [19] and BPOSS-PDI-OH [61], respectively. The former removes the influence of PDI aggregation and the latter separates the effect of C_60_. For BPOSS-C_60_, the major form is the dimeric species (namely, dimers **1**-**4**) and little monomeric and oligomeric species. The dissociation energy for the most stable dimer **4** of BPOSS-C_60_ is measured to be ~0.66 eV (Figure 3), which is very close to the energy at which the dimer **3** of BPOSS-PDI-C_60_ dissociates (Figure 2B). Therefore, it is inferred that dimer **3** of BPOSS-PDI-C_60_ results from π–π interaction between the C_60_ motifs and that dimers **1** and **2** of BPOSS-PDI-C_60_ may be caused by the interaction between C_60_ and PDI positioned in different ways [62]. For BPOSS-PDI-OH (Appendix A), the dimers **2** and **3** are the most abundant and the most stable. They are probably attributed to the longitudinally and transversely offset in the π–π stacking of PDI planes to make the dimer more compact. An interesting observation is that dimer **4** of BPOSS-PDI-OH shows low intensity but extremely high stability. Since the drift time of dimer **4** is longer than the other three dimeric species, its conformation is expected to be less compact. It is suggested that the formation of dimer **4** is caused by longitudinal displacement and further stabilized by hydrogen bonding [61].

### 2.3. Solution Assembly as Revealed by UV/Vis and Fluorescence Spectrometry

The BPOSS-PDI-OH contains one hydroxyl group and one BPOSS motif thus, its solubility in common organic solvents such as CH_2_Cl_2_, CHCl_3_ and THF are significantly enhanced. The absorption spectra of BPOSS-PDI-OH in CHCl_3_ at different concentrations are shown in Figure 4A. The UV/Vis spectra display the typical pattern for the monomeric form of PDI with characteristic peaks at 460 nm, 492 nm, and 528 nm, corresponding to the 0–2, 0–1, and 0–0 electronic transitions, respectively [63,64,65].

The 0–1 transition changed slightly after normalization implies that the steric effect of the POSS group decreases the aggregation of the PDI cores [47]. However, upon increasing concentration, the ratio of the absorbance at 492 nm and 528 nm (*A*_492_/*A*_528_) shows a transition in slope at ~1.1 × 10^−4^ M (Figure 4B), which suggests that there may be a change in the aggregation states. Above this critical concentration, BPOSS-PDI-OH may exist as an equilibrium between dimer and different forms of higher aggregates. At even higher concentrations, the absorbance would go beyond the instrument’s limit and the spectra would be deformed. Meanwhile, precipitation would also occur which results would not be reliable.

Photoluminescence is also sensitive to the aggregation states. The fluorescence spectra of BPOSS-PDI-OH in CHCl_3_ with increasing concentration (1 × 10^−7^~1 × 10^−4^ M) are depicted in Figure 4C,D. Generally speaking, PDI has a very high fluorescence quantum yield approaching unity in dilute solution [46]. At low concentrations, the fluorescence intensity increases with increasing concentration (Figure 4C). At concentrations higher than 2 × 10^−5^ M, the trend is changed (Figure 4D). The fluorescence intensity is gradually quenched with increasing concentration, which is probably due to the formation of dimers and other aggregates that causes partial quenching, a typical phenomenon which is known as aggregation-caused quenching. The transition concentration was found to be slightly lower than that found in UV/Vis absorption spectra, which is reasonable since fluorescence is usually more sensitive. As revealed previously, steric hindrance plays a key role in disrupting the strong, continuous π–π stacking and limit the aggregation to discrete dimers, even at high concentrations. This is supported by the only slightly red-shifted excimer-like fluorescence in CHCl_3_ upon increasing concentration, which indicates the formation of ground state complexes with limited π–π stacking interaction in solution.

Unlike most PDI or C_60_ derivatives, BPOSS-PDI-C_60_ is quite soluble in CHCl_3_ which facilitates processing and further self-assembly studies. Figure 5A shows the UV/Vis absorption spectra of the compound at various concentrations in CHCl_3_. It is essentially the overlay of the UV/Vis absorption spectra of the two chromophores (PDI and C_60_). Typical absorption patterns for PDI monomers could be observed with absorbance in the order of *A*_462_ < *A*_492_ < *A*_528_. Meanwhile, the absorption of C_60_ could also be observed in the shorter wavelength region. Upon increasing the concentration, the absorbance increased gradually and there is neither new peak formation nor peak position shifting. However, the relative peak intensities did change, indicating a change in their aggregation states. From the plot of the ratio of absorbance at 492 nm and 528 nm (*A*_492_/*A*_528_), we could clearly identify a transition in slope at ~5.6 × 10^−5^ M (Figure 5B). The value is about half that of BPOSS-PDI-OH, which is reasonable considering the lower solubility of BPOSS-PDI-C_60_. With the introduction of C_60_, the fluorescence intensity of PDI is dramatically reduced as compared to that of BPOSS-PDI-OH (Figure 5C), indicating strong quenching of the excited singlet state of the PDI moiety by C_60_ due to their electronic interactions. The observation is consistent with previous reports on PDI-C_60_ conjugates [66]. Nonetheless, the concentration-dependent fluorescence spectra still reveal some important information about the aggregates (Figure 5D). First, at low concentrations, there is an increase in fluorescence intensity with increasing concentration, but there is neither new peak formation at longer wavelength, nor peak shifting up to 1 × 10^−5^ M. By further increasing the concentration, there is a shift in fluorescent peak wavelength as well as a change in relative peak intensities. The aggregation-caused quenching phenomenon is similar to that of BPOSS-PDI-OH.

Generally speaking, there are mainly two types of aggregates in conjugated chromophores: the H-type and the J-type. H-aggregates are evidenced when the absorption maximum of the aggregates is blue-shifted with respect to the isolated chromophore and the fluorescence yield is lowered, whereas J-aggregates occur if absorption and emission maxima are red-shifted without fluorescence quenching. In both cases, continuous packing of the chromophore is implicated. By looking at the concentration-dependent absorption spectra and the corresponding fluorescence spectra, it is hard to draw a conclusion on which type of aggregates was actually formed. There was essentially no shifting of peak positions and intensities. It indicates that both BPOSS-PDI-OH and BPOSS-PDI-C_60_ might form unique discrete aggregate unlike traditional H- or J-type aggregates. This is consistent with previous results on BPOSS-PDI-BPOSS in that such compounds most likely would prefer forming discrete dimers as the packing motif in the aggregates. The large steric hindrance from the bulky molecular nanoparticles likely disrupts the continuous packing and limits the number of molecules in one aggregate [46]. By removing one POSS unit, more degree of freedom is allowed, leading to more variation in the type of dimers as observed in the TWIM-MS spectrum of BPOSS-PDI-OH. The addition of C_60_ introduces strong interaction between C_60_, which might override other interactions and lead to a predominant form of the stable dimer. In both cases, the bulky BPOSS is expected to play an important role in restraining the strong π–π interaction of PDIs.

### 2.4. Self-Assembly of BPOSS-PDI-OH into Single Crystalline Nanobelts

In previous sections, we have carefully looked into the self-assembly of BPOSS-PDI-OH and BPOSS-PDI-C_60_ both in gas phase and in solution. The assembly in gas phase reveals the most favored interaction pattern between these molecules. In this case, both shape amphiphiles favor the formation of dimers because π–π interaction is the strongest interaction in this system and the presence of bulky POSS moiety prevents extensive interaction and continuous stacking. The assembly in solution largely follows the same trend with dimer being the most favored motif of aggregation with increasing concentration. This is evidenced by the unique spectral changes upon aggregation that could not be attributed to either H-type or J-type aggregates. It is thus intriguing to further explore their self-assembly in solid state, especially in crystals. One would anticipate that dimers could also serve as the building block in solid state for further packing into crystals, just like that observed in BPOSS-PDI-BPOSS derivatives [46,50,67]. Quantum mechanics studies will also be greatly helpful to further understand the molecular interactions and connect the structural-property relationship [68]. Meanwhile, since single crystal engineering has shown great potential for creating tailored structures for application in microelectronics, it attracts considerable interest to grow crystals from these shape amphiphiles and to take advantage of their shape disparity and create novel structures with periodically arranged PDI and C_60_ motifs. Hence, we tried to grow single crystals of both compounds.

The first attempt was made following the two-phase slow diffusion method reported for growing the BPOSS-PDI-BPOSS single crystals [67]. However, it only affords polycrystalline powders rather than single crystals. The slow evaporation method was also tried using a dilute solution (see Appendix A for details). The solvent evaporation rate was controlled to be very slow so that the single crystal growth on various substrates like carbon-coated mica sheet could be ideal. A chamber saturated with THF atmosphere was used to control the evaporation rate. Various conditions, such as solution concentration, choice of solvents, the evaporation rate, etc., are optimized, which is critical for success. Indeed, single crystals of BPOSS-PDI-OH could be grown on the mica sheet with the overall shape resembling nanobelts. Figure 6A,B show well-defined, flat nanobelt-shaped single crystals with length as long as several hundred micrometers. In order to confirm the single crystallinity of the nanobelt, TEM and SAED experiments were also carried out. Perfect nanobelt-shaped crystals can be clearly seen under TEM and an excellent single crystal diffraction pattern could be observed (Figure 6C,D). Unfortunately, we were unable to grow single crystals of BPOSS-PDI-C_60_ after numerous trials, which could probably be explained by the strong and unbalanced interactions between various molecular moieties. Only amorphous powder was obtained, as suggested by the XRD pattern (Appendix A). The limitations imposed on our research by present facilities prevent us from the further study on their electronic properties, which are intriguing topics of future research.

## 3. Materials and Methods

### 3.1. Materials and Instruments

Aminophenylisobutyl POSS (BPOSS-NH_2_, AM0292, ≥97.0%, Hybrid Plastics, Hattiesburg, MS, USA), (60)fullerene (C_60_, MTR Ltd., 99.5%), perylene-3,4,9,10-tetracarboxylic dianhydride (PTCDA, ≥98.0%, Aldrich, Saint Louis, MS, USA), 4-dimethylaminopyridine (DMAP, 99.0%, Aldrich), *N*,*N*′-diisopropylcarbodiimide (DIPC, 98.0%, Aldrich), imidazole (ACS reagent, ≥99.0%, Aldrich), 3-amino-1-propanol (98.5%, J&K Chemical, Wenzhou, Zhejiang province, China), 1,2-dichlorobenzene (ODCB, anhydrous, 99.0%, Aldrich), chloroform (ACS reagent, ≥99.8%, Aldrich), and tetrahydrofuran (THF, anhydrous and inhibitor-free, ≥99.9%, Aldrich) were employed. All reagents were used as received without further purification. The malonic acid derivative of C_60_ was prepared according to literature [69].

NMR spectra were recorded on a Varian 500 MHz spectrometer (Varian Inc. Palo Alto, CA, USA) for ^1^H-NMR and 125 MHz for ^13^C-NMR. Chemical shifts (δ values) were reported in ppm, residual resonances from CDCl_3_ was set as 7.27 ppm for ^1^H NMR and 77.00 ppm for ^13^C NMR. Infrared spectra were recorded on an Excalibur Series FT-IR spectrometer (DIGILAB, Randolph, MA, USA). Matrix-assisted laser desorption/ionization time-of-flight (MALDI-TOF) mass spectra were recorded on a Bruker Reflex-III TOF/TOF mass spectrometer (Bruker Daltonics, Billerica, MA, USA). Differential scanning calorimeter (DSC, Perkin-Elmer PYRIS Diamond with Intracooler 2P cooling system, CA, USA) experiments were carried out to observe possible phase transitions. The thermogravimetric analysis (TGA) of the samples was analyzed with a TA Instrument-Water LLC Q500 (New Castle, DE, USA). One-dimensional (1D) wide angle X-ray diffraction (WAXD) experiments were conducted using a Rigaku Multiflex 2 kW automated diffractometer using Cu Kα radiation (0.1542 nm) in reflection mode (Rigaku Corporation, Tokyo, Japan). Transmission electron microscopy was performed on Philips Tecnai 12 at an accelerating voltage of 120 kV (FEI Company, Hillsboro, OR, USA). UV/Vis spectra were recorded on a Lambda 35 (Perkin Elmer) spectrophotometer. Electrospray ionization (ESI) mass spectra were obtained on a Waters Synapt HDMS quadrupole/time-of-flight (Q/ToF) tandem mass spectrometer equipped with traveling wave ion mobility (TWIM) separation (Waters Corporation, MA, USA).

### 3.2. Synthesis

#### 3.2.1. Synthesis of BPOSS-PDI-OH

POSS-monotethered perylene diimide with flexible linker (BPOSS-PDI-OH) was synthesized following the standard condensation method developed by Langhals (Scheme 1) [70]. A modified one-step procedure was used in this synthesis [56]. BPOSS-NH_2_ (0.36 g, 1.1 mmol), PTCDA (0.72 g, 1.85 mmol) and imidazole were dissolved in *o*-dichlorobenzene (ODCB) and placed in a nitrogen-purged flask fitted with a reflux condenser. The mixture was heated in an oil bath at 140 °C under N_2_, then BPOSS-NH_2_ (0.36 g, 1.1 mmol, dissolved in THF) and 3-amino-1-propanol (0.39g, 5.1 mmol, dissolved in THF) were added separately using a syringe pump under vigorous stirring for 3 h. Then the reaction mixture was cooled to room temperature and rinsed in 50 mL of ethanol followed by the addition of 200 mL of 2 N HCl. The mixture was extracted with 200 mL CHCl_3_ two times after stirring overnight. The combined organic phase was washed thoroughly with distilled water until the pH value of washings turned to be neutral. After drying over anhydrous sodium sulfate, the solvent was concentrated and the crude sample was loaded into a silica gel column and eluted with hexane/ethyl acetate (*v*/*v* =10:1). The colored fraction was collected, which was a mixture of the desired product and unreacted raw material. It was further purified by dissolving in chloroform and adding methanol to induce the precipitation of BPOSS-PDI-OH by slow diffusion. The product was collected by filtration and obtained as a red solid in ~30% yield. ^1^H-NMR (CDCl_3_, 500 MHz, ppm, δ): 8.69–8.56 (m, 8H), 4.41 (t, *J* = 10, 2H), 4.21 (t, *J* = 10, 2H), 3.66 (dt, *J*_1_ = 10, *J*_2_ = 10, 2H), 3.04 (t, *J* = 10, 2H), 2.10–2.02 (m, 2H), 1.91–1.78 (m, 9H), 0.96–0.93 (m, 42H), 0.78–0.72 (m, 2H), 0.62–0.57 (m, 14H). ^13^C-NMR (CDCl_3_, 125 MHz, ppm, δ): 163.7, 162.7, 134.5, 133.7, 131.3, 130.9, 129.1, 128.9, 125.9, 125.8, 123.4, 122.9, 122.7, 59.2, 54.0, 53.9, 42.9, 37.2, 31.1, 31.0, 30.1, 30.0, 25.6, 25.0, 23.6, 23.4, 21.7, 9.9. FT-IR (KBr) wavenumber (cm^-1^): 3480, 2954, 2867, 1701, 1661, 1596, 1348, 1228, 1111, 744. MALDI-TOF MS (*m*/*z*): Calcd for NaC_58_H_84_N_2_O_17_Si_8_ (M∙Na^+^): 1327.4; Found: 1327.6.

#### 3.2.2. Synthesis of BPOSS-PDI-C_60_

To a solution of carboxylic acid derivative of methanofullerene C_60_-COOH (58.4 mg, 0.08 mmol) in 3 mL of ODCB/ DMF mixed solvent (*v*/*v* 15/1), BPOSS-PDI-OH (100 mg, 0.08 mmol) and DMAP (1.7 mg, 0.50 mmol) in 8 mL toluene were added followed by *N*,*N*′-diisopropylcarbodiimide (DIPC, 13.18 mg, 1.03 mmol). The mixture was stirred at room temperature for 24 h. After that, the solution was washed with brine (10 mL) and H_2_O (10 mL). The organic phase was dried over MgSO_4_, concentrated, and purified by silica gel column chromatography using hexane/toluene (*v*/*v* = 2/1) as eluent to give BPOSS-PDI-C_60_ as a dark brown powder (136.2 mg; Yield: 87%). ^1^H NMR (300 MHz, CDCl_3_, ppm, *δ*): 8.76–8.55 (m, 8H), 4.67–4.59 (dt, *J*_1_ = 6, *J*_2_ = 6, 4H) 4.28 (s, 1H), 4.22 (t, *J* = 6, 2H), 2.49–2.41 (m, 2H), 1.92–1.79 (m, 9H), 0.96–0.93 (m, 42H), 0.77–0.71 (m, 2H), 0.63–0.59 (m, 14H). ^13^C NMR (75 MHz, CDCl_3_, ppm, *δ*): 164.9, 162.6, 162.0, 146.9, 144.5, 144.3, 144.2, 144.1, 144.0, 143.8, 143.7, 143.6, 143.6, 143.6, 143.3, 143.2, 143.2, 142.9, 142.5, 142.2, 142.0, 141.9, 141.9, 141.7, 141.6, 141.1, 141.0, 141.0, 140.9, 139.8, 139.7, 139.4, 134.9, 133.9, 133.2, 130.6, 130.2, 128.4, 128.2, 125.5, 122.5, 122.2, 122.0, 121.9, 69.3, 63.8, 41.9, 37.7, 37.1, 29.3, 28.7, 26.1, 24.7, 24.7, 24.4, 22.9, 20.8, 21.5, 21.4, 20.5, 8.8. FT-IR (KBr) wavenumber (cm^−1^): 2953, 2870, 1743, 1699, 1662, 1595, 1443, 1344, 1230, 1101, 839, 812, 744, 484. MALDI-TOF MS: Calcd. for NaC_120_H_84_N_2_O_18_Si_8_, 2087.4, Found: 2087.7.

## 4. Conclusions

In summary, two shape amphiphiles, BPOSS-PDI-OH and BPOSS-PDI-C_60_, were synthesized and fully characterized. The ESI-TWIM-MS spectrometry was utilized to characterize the aggregates formed in gas phase. It was found that they show distinct dimeric packing schemes owing to the presence of different interactions between distinct molecular moieties. The hydroxyl group seems to elicit strong hydrogen bonding to stabilize the otherwise unstable forms of aggregates. Even π–π interaction could have significant differences due to the anisotropy in molecular shape, as in C_60_ and PDI. The UV/Vis and fluorescence emission spectrometry demonstrated concentration-dependent self-assembly for both samples. Furthermore, nanobelt-shaped single crystals of BPOSS-PDI-OH with very regular morphology could be obtained that are hundreds of micrometers in length and several hundreds of nanometers in width. The study shows that the incorporation of multiple molecular entities of distinct shape and interaction patterns would dramatically complicate the self-assembly and lead to unique nanostructures with intriguing photo- and electro- properties for various applications.

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
