# Peer review of "Synthesis, Self-Assembly and Characterization of Tandem Triblock BPOSS-PDI-X Shape Amphiphiles"

_molecules, 2019, doi:10.3390/molecules24112114_

Round 1
Reviewer 1 Report
This article “Synthesis, Self-Assembly and Characterization of Tandem Triblock BPOSS-PDI-X Shape Amphiphiles” by Shao et al. is well-written and highly relevant to the readership of Molecules. The authors report a straightforward approach for synthesizing shape amphiphiles with functional moieties. The chemical identification uses the typical suite of measurements such as NMR, IR, and MS, while further microscopy and spectroscopy methods are used to understand how the material aggregates in different physical conditions. I recommend publishing this manuscript after a couple of minor revisions.
Figure 4c-d: Regarding the solution-based aggregation, do the authors suggest that the UV-vis spectra represent equilibrium aggregates? Or is further aggregation expected, but with slower kinetics? One suggestion to resolve this question is to monitor the UV-vis spectrum of a given concentration versus time. This observation has implications on the stability BPOSS-PDI-OH aggregates.
Line 299: “Various conditions, such as solution concentration, choice of solvents, the evaporation rate, etc., are optimized, which is critical for the success.” It is advantageous to state the parameters that successfully produced pristine single crystals.
Author Response
Response to Reviewer 1 Comments
This article “Synthesis, Self-Assembly and Characterization of Tandem Triblock BPOSS-PDI-X Shape Amphiphiles” by Shao et al. is well-written and highly relevant to the readership of Molecules. The authors report a straightforward approach for synthesizing shape amphiphiles with functional moieties. The chemical identification uses the typical suite of measurements such as NMR, IR, and MS, while further microscopy and spectroscopy methods are used to understand how the material aggregates in different physical conditions. I recommend publishing this manuscript after a couple of minor revisions.
Point 1: Figure 4c-d: Regarding the solution-based aggregation, do the authors suggest that the UV-vis spectra represent equilibrium aggregates? Or is further aggregation expected, but with slower kinetics? One suggestion to resolve this question is to monitor the UV-vis spectrum of a given concentration versus time. This observation has implications on the stability BPOSS-PDI-OH aggregates.
Response 1: We thank the reviewer for the critical comments. In the current study, we focused on the equilibrium states under different concentrations, solutions were allowed to stand overnight before spectra were taken. To be more specific, we add “Sample solutions with different concentrations are put still overnight to reach the equilibrium state and confirmed no precipitates before testing.” in the supporting information of UV-vis testing.
Point 2: Line 299: “Various conditions, such as solution concentration, choice of solvents, the evaporation rate, etc., are optimized, which is critical for the success.” It is advantageous to state the parameters that successfully produced pristine single crystals.
Response 2: We thank the reviewer for the thoughtful comments. The parameters that successfully produced pristine single crystal is in the Experimental Section of supporting information named Single Crystal Growth. The description was copied as “The single crystals of BPOSS-PDI-OH were obtained by slow evaporation from dilute solution on substrates in a solvent-saturated atmosphere. A chamber with a THF saturated atmosphere was constructed using a culture dish with a steel cylindrical mount at the bottom and 1 ml of THF at the bottom of the dish. A square substrate of about 1.0×1.0 cm2 (e.g. carbon-coated mica) was placed on the top of the mount and the dish was sealed with a glass cover. The sealed chamber was kept at room temperature for 1h for the atmosphere to reach THF saturation. The solvent was allowed to slowly evaporate through the interface between the glass cover and the dish wall. A micro-syringe was used to deposit one drop of the dilute solution on the substrate. The chamber was then left to allow for the solvent to completely evaporate. The single crystals on the substrate were examined by optical microscopy (Olympus BX52) and then collected for characterization using a variety of techniques.”
We apologize for not providing the detailed concentration, evaporation rate, etc. to avoid some misleading, because the single crystal growth is strongly dependent case by case. The key issue, in this case, is the saturated THF atmosphere.

Reviewer 2 Report
Authors have synthesized BPOSS-PDI-OH and BPOSS-PDI-C60 shape amphiphiles and studied their self-assembly.
1. Authors have mainly focused on the π- π stacking in the discussion part as a driving non-covalent interaction but the conclusion includes H-bonding also. Please discuss the possible role of hydrogen bonding in the self-assembly process.
2. Apart from steric hindrance caused by fullerene, what other forces do you think are involved here.
3. These papers can be cited:
P. Politzer, J. S. Murray, Crystals. 9, 165 (2019).,P. K. Mishra, A. Ekielski, Nanomaterials. 9, 243 (2019).,
Author Response
Response to Reviewer 2 Comments
Authors have synthesized BPOSS-PDI-OH and BPOSS-PDI-C60 shape amphiphiles and studied their self-assembly.
Point 1: Authors have mainly focused on the π- π stacking in the discussion part as a driving non-covalent interaction but the conclusion includes H-bonding also. Please discuss the possible role of hydrogen bonding in the self-assembly process.
Response 1: We thank the reviewer for the critical comments. In this study, we focused on the π-π stacking and consider it is the major non-covalent bond leading the self-assembly. However, the H-bonding will indeed be a nontrivial interaction in molecular packing. Because of the existence of C=O and -OH groups, there will be mainly two kinds of H-bonding in the self-assembly structure: 1) -OH and a carbonyl group on the PDI core, and 2) -OH between two BPOSS-PDI-OH. Since the detailed investigations of BPOSS-PDI-OH and BPOSS-PDI-OCH3 have been published elsewhere and citied (Ref 60: Angew. Chem. Int. Ed. 2017, 56, (6), 1452-1464.), we feel it will be a redundancy to go detailed discussion.
Point 2: Apart from steric hindrance caused by fullerene, what other forces do you think are involved here.
Response 2: We thank the reviewer for the thoughtful comments. Apart from the forces between covalent bonds, there will be lots of noncovalent interactions such as Coulombic force, hydrogen bonding, van der Waals force, etc. which will greatly influence the molecular behavior. In the current case, we compared the dissociation energy between BPOSS-PDI-C60, BPOSS-C60 and other PDI derivatives with no bulky groups. We found the planar π-π stacking of PDI is strongly hindered by BPOSS or C60 motif. However, we can’t quantify this hindrance or other forces such as the aggregation of C60. Unfortunately, this is phenomenal research and we hope further studies can be carried out by quantum mechanics to make a better connection between theory and experiments.
3. These papers can be cited: P. Politzer, J. S. Murray, Crystals. 9, 165 (2019).,P. K. Mishra, A. Ekielski, Nanomaterials. 9, 243 (2019).,
Response 3: We thank the reviewer for the kind suggestion. “Crystals.9, 165” was cited at Page 2, Line 88 as ref. 53 and Nanomaterials.9, 243 was cited at Page 9, Line 288 as “Quantum mechanics studies will also be greatly helpful to further understand the molecular interactions and connect the structural-property relationship[68].”
